

# Cosmic-ray neutron method for the continuous measurement of Arctic snow accumulation and melt

Anton Jitnikovitch[1], Philip Marsh[1], Branden Walker[1] and Darin Desilets[2].

5  [1]Cold Regions Research Centre

Wilfrid Laurier University

75 University Avenue

Waterloo, ON, N2L 3C5, Canada

10  [2]Hydroinnova LLC

1401 Morningside Drive NE

Albuquerque, NM, 87110, USA

*Correspondence to:*  Anton Jitnikovitch (email: antonjitnikovitch@hotmail.com)

**Abstract.** The Arctic is warming at two to three times the rate of the global average, significantly impacting snow accumulation and melt. Unfortunately, conventional methods to measure snow water equivalent (SWE), a key aspect of the Arctic snow cover, have numerous limitations that hinder our ability to document annual cycles, the impact of climate change, or to test predictive models. As a result, there is an urgent need for improved methods that measure Arctic SWE; allow for continuous, unmanned measurements over the entire winter; and allow measurements that are representative of spatially variable, Arctic snow covers. In-situ, or invasive, cosmic ray neutron sensors 20  (CRNSs) may fill this observational gap, but few studies have tested these types of sensors or considered their applicability at remote sites in the Arctic. During the winters of 2016/17 and 2017/18 we tested an in-situ CRNS system at two locations in Canada; a cold, low- to high-SWE environment in the Canadian Arctic and at a warm, low-SWE landscape in Southern Ontario that allowed easier access for validation purposes. CRNS moderated neutron counts were compared to manual snow survey SWE values obtained during both winter seasons. Pearson correlation coefficients ranged from -0.89 to -0.98, while regression analyses provided $R^2$ values from 0.79 to 0.96. RMSE of the CRNS-25  measured SWE averaged 2 mm at the southern Ontario site and ranged from 28 to 40 mm at the Arctic site. These data show that in-situ CRNS instruments are able to continuously measure SWE with sufficient accuracy, and have important applications for measuring SWE in a variety of environments, including remote Arctic locations. These sensors can provide important SWE data for testing snow and hydrological models, water resource management applications, and the validation of remote-sensing applications.

Keywords: Arctic, snow water equivalent, cosmic ray sensors, neutrons



## 1. Introduction

The Arctic tundra snow cover is typified by low snow depth and low snow water equivalent (SWE) when averaged over areas of a few km$^2$, but extreme spatial variability in depth and SWE over distances of less than 10 m (Sturm et, 1995, 2001; Rees et al., 2014). These features are due to a combination of low winter snowfall, wind that redistributes snow across

the landscape, and high rates of sublimation during these blowing snow events. For example, snowfall is often less than 300 mm over the long winter, with up to 40% of this snowfall sublimating during blowing snow events. Blowing snow also results in wind scoured uplands characterized by shallow, low density snow cover (< 0.7 m; < 300 kg/m$^3$) and deep, high density snow drifts (up to 10 m; up to 600 kg/m$^3$) located on steep hillslopes (Marsh and Pomeroy, 1996). Within the tundra-taiga ecotone, deep drifts also occur in small shrub or tree patches. Although deep drifts are small in area, they often contain a large

portion of the total landscape SWE (Gray et al., 1974; Marsh and Woo, 1981; Gray et al. 1989; Marsh and Pomeroy, 1996; Sturm et al., 2001). This spatially variable snow cover exerts important controls on many aspects of the tundra environment, including soil and permafrost temperature, permafrost processes such as ice wedge cracking, streamflow hydrology, lake level, and wildlife habitat for example. However, monitoring this snow cover remains extremely challenging (Kinar and Pomeroy, 2015).

The Arctic snow observing system has very few ground-based monitoring stations, and these are often located at non-representative locations (Schiermeier, 2006; Rees et al., 2014; Goodsite et al., 2016). In addition, standard measurements used at these stations are either prone to large errors, not representative of the surrounding area, or not measured at all. For example, snowfall measurements are prone to large errors due to under catch during high winds (Pan et al., 2016), while sublimation is seldom measured. Measurements of snow depth are typically not representative of the surrounding natural terrain as they are

limited to point observations using ruler measurements or acoustic distance systems (Kinar and Pomeroy, 2015). Recent advances in methodology allow measurement of SWE using gamma attenuation or global positioning systems (Koch et al., 2019) for example, but again are limited to point measurements. To overcome these deficiencies, practitioners and researchers still use traditional, manual snow surveys in order to document average snow depth, density and SWE across Arctic landscapes. Snow survey methods have well known limited accuracy in tundra areas (Goodison et al., 1981; Pomeroy and Gray, 1995;

Steufer et al., 2013; Kinar and Pomeroy, 2015) and do not allow for mapping snow cover as is needed for many Arctic research studies.

Satellite and aircraft remote sensing provide methods to partially overcome some of the limitations outlined above through the mapping of both snow cover extent and SWE. Although current satellite methods are well suited to assessing climate change impacts on snow across the entire Arctic (Derksen and Brown, 2012; Rees et al., 2014; Hori et al., 2017; Tollefson, 2017; Bush and Lemmen, 2019) and for large scale water resource needs, they are not suited for providing snow data at the high spatial resolution needed for many research needs. Airborne remote sensing methods are able to provide high resolution snow data, but also have certain limitations. For example, methods to map snow depth at high resolutions are available (Deems et al, 2013; Walker et al., 2020), but mapping of snow density or SWE are not (Koch et al., 2019). SWE along flight transects is available using airborne gamma methods but have limited applicability in the Arctic due to high cost. Airborne radar methods have promise for mapping SWE at moderate resolution but remain in the research stage (Derksen et al., 2017).

Cosmic ray attenuation methods have not been extensively tested but may fill a needed gap between existing ground-based and remote sensing snow monitoring methods. Kodama et al. (1979) first described the use of an in-situ, or invasive, cosmic ray neutron sensor (CRNS) to measure SWE by burying a shielded neutron sensor below the ground surface and allowing snow to accumulate upon it. This method records neutrons in the fast to epithermal range which are generated by galactic cosmic rays that interact with atmospheric particles, snow and soil (Kodama et al., 1979; Howat et al., 2018; Gugerli et al., 2019). As hydrogen in water molecules absorbs neutrons, higher SWE snowpacks will attenuate larger numbers of neutrons, leading to lower neutron counts below deeper snow packs. For a neutron sensor placed at ground level, the sensor footprint is essentially a point source on the scale of the instrument (in our case, a 130 cm tube), and the relationship between neutron counts is inversely proportional to the amount of SWE on the ground. Currently, we are aware of two such in-situ CRNS systems that are used operationally or are commercially available. One is deployed by Électricité de France (Paquet and Laval, 2005; Paquet et al., 2008; Delunel et al., 2014) in the French Alps and used in estimating snow cover runoff for operational hydroelectric power generation. A second CRNS system is the SnowFox[TM] (SF) system commercially available

from Hydroinnova (Howat et al., 2018; Gugerli et al., 2019). The SF uses a single neutron measuring tube placed immediately

below or at the ground surface prior to winter, allowing the snowpack to accumulate atop of it. Hydroinnova also produces the

CRS-1000™, a non-invasive CRNS, which uses two neutron measuring tubes located above the ground and snow surfaces.

With sensor above the snow surface, the footprint of the system is increased substantially. Although there remains some

uncertainty in the size of the footprint, it may be as high as 300 m radius around the sensor (Desilets et al., 2010; Zreda et al.,

2012; Desilets and Zreda, 2013; Köhli et al., 2015; Sigouin and Si, 2016; Schattan et al. 2017). Further work is required on

testing such a system in the Arctic.

The CRS-1000 and SF potentially complement each other for measuring snow, with the CRS-1000 providing an

average SWE over a large area surrounding the sensor, and the SF measuring point SWE along a transect. Since the SF can

measure SWE from near zero to as high as four meters, and potentially up to 10 meters (Howat et al., 2018; Gugerli et al.,

2019), the SF could potentially measure SWE across deep snow drifts. Such a network of CRNS sensors has the potential to

fill a significant measurement gap between traditional ground-based measurement systems and remote sensing. This paper will

focus on the in-situ type of CRNS system, the SF, simply called a CRNS for the remainder of the paper, with an objective to

test the potential of this CRNS to provide continuous measurements of SWE accumulation and melt along transects where

SWE varies greatly, and over full snow seasons. Future research will focus on the CRS-1000 sensor.

## 2. Materials and Methods

### 2.1 Cosmic Ray Neutron Sensor (CRNS)

The CRNS has a single neutron sensor tube, installed on the ground surface, that provides an estimate of SWE across

a small footprint that is assumed to be a "point" measurement. The CRNS used in this study has a 130 cm cylindrical neutron

detector tube with a separate control module incorporating a Hydroinnova QDL2100 data logger and an iridium satellite

communication device. The neutron detector tube is moderated (shielded) by a polyethylene casing to reduce the sensitivity of

the detector gas and to increase the sensitivity towards the fast and epithermal ranges – where the CRNS principally measures

neutrons after they traverse the overlying snowpack (Delunel et al., 2014; Woolf et al., 2019). Between this energy range, a

neutron collision with the CRNS polyethylene casing causes the neutron to reach thermal equilibrium with the moderator and be easily absorbed by the detector. An absorber in the detector tube captures the neutron and splits into two charged particles which trigger an ionization pulse in the tube, this is noted as one neutron count (Bartol, 1999). Counts are recorded over a pre-

set interval and the counting rate (i.e. relative neutron intensity) can be retrieved manually from the data logger and are also posted in near real-time on a private web portal hosted by the manufacturer. The fundamental process of the CRNS is that a baseline moderated neutron counting rate is established during the initial snow-free setup, and any deviations from this baseline would be inversely proportional to the amount of near-surface water content. This near-surface water content is primarily attributed to SWE during snow covered periods, and to soil moisture during snow-free periods. A single neutron tube can be

used individually, or a number of neutron sensor tubes can be connected to a single data logger to provide measurements along a transect up to several hundred meters in length. Due to the fundamental operation of the CRNS, when setting up multiple neutron sensor tubes in a transect, it is strongly recommended that an identical moderated neutron counting rate is used as the baseline for each unit.

**2.2 Determination of Snow Water Equivalent using a Cosmic Ray System**

115         To estimate SWE from the CRNS neutron data, the raw moderated neutron counts ($N_{RAW}$) must be corrected for barometric pressure ($F_p$) and the temporal variation of incoming neutrons ($F_i$). Since these correction factors ($F_p$ and $F_i$) represent a change from one point in time to another, they are unitless. The corrected moderated neutron counts (N) is calculated as:

$$N = N_{RAW} \times F_p \times F_i \tag{1}$$

N is then updated as a running average over 12 timesteps in order to reduce the noise associated with the hourly moderated neutron data. $F_p$ is given by:



$$F_p = exp\left(\frac{P-P_0}{L}\right) \tag{2}$$

where $exp$ is the natural exponential, P is the observed air pressure (hPa) recorded by a pressure sensor on the CRNS

instrument, and $P_0$ represents a reference air pressure, set to 1000 hPa. The mass attenuation length, L (g/cm$^2$), was provided

by the manufacturer and is based on latitude (Desilets, 2021). $F_i$ is then calculated as:


$$F_i = \frac{N_{ref}}{N_{nm}} \tag{3}$$

where $N_{ref}$ is the average incoming neutron count over an arbitrary counting period (e.g. the first month of data after the initial

snow-precipitation of the winter season) and $N_{nm}$ is the hourly incoming neutron count during the time of interest. Numerous

non-invasive CRNS studies (Zreda et al., 2012; Chrisman and Zreda, 2013; Schattan et al. 2017; Schattan et al., 2019) have

used incoming cosmic ray fluxes from the Jungfraujoch Neutron Monitor in Switzerland to estimate $F_i$. However, incoming

cosmic rays are location dependent, and neutron monitoring stations with higher geomagnetic latitudes are known to have a

greater sensitivity to the lower end of the neutron monitor energy range, when compared to midlatitude or low-latitude stations

(Kuwabara et al., 2006). As a result, it is preferable to use a nearby neutron monitor, and we therefore use incoming neutron

fluxes from the monitoring station located at the Aurora Research Institute, Inuvik, Northwest Territories, and available from

the Neutron Monitor Database (Klein et al., 2010). SWE (mm) can then be estimated as follows (Desilets, 2010):

$$SWE = -10 \times (\varLambda) \times \ln\left(\frac{N}{N0}\right) \tag{4}$$

where ln is the natural logarithm, N is the corrected and 12-h averaged moderated neutron count from Eq. (1), and $N_0$ represents

the averaged neutron count 7-14 days prior to the initial snow accumulation of the season. $N_0$ serves as the instrument's

moderated neutron count baseline, establishing a crucial initial relationship between the pre-snowfall neutron count and a near-





surface water content while the SWE is zero. Any deviations from the baseline counting rate are inversely proportional to the

amount of near-surface water content. This is the fundamental operating process of the CRNS instrument. The near-surface

water content range for this in-situ CRNS has not been quantified in literature, however, it is primarily attributed to SWE

during snow covered periods and soil moisture during snow-free periods (Paquet and Laval, 2005; Paquet et al., 2008; Howat

et al, 2018). The attenuation coefficient, $\frac{1}{\Lambda}$ , is then calculated as:

$$\frac{1}{\Lambda} = \frac{1}{\Lambda_{max}} + \frac{\left(\frac{1}{\Lambda_{min}} - \frac{1}{\Lambda_{max}}\right)}{\left(1 + exp\left(-\frac{\left(\frac{N-a_1}{N_0}\right)}{a_2}\right)\right)^{a_3}} \qquad (5)$$


For details regarding the CRNS parameters, refer to Sect. 3.4.

## 3. Study Sites and Methods

### 3.1 Study Sites

CRNSs were installed at two locations across Canada; a warm, low SWE agricultural field located in southern Ontario,

and a cold, high SWE environment located within a tundra shrub patch in the western Canadian Arctic (Fig. 1). The southern

site allowed frequent field visits during the winter period, and the combination of two sites allowed testing of the CRNS over

a range of SWE, climate, and soil conditions. The southern Ontario study site is located at 300 masl, near Elora, Ontario (43.6°

N, 80.3° W) (Fig. 1). This site typically has warm, shallow snowpacks with low SWE and low spatial variability. A dominant

feature of the Elora site is the absence of a consistent average annual snowpack, numerous snowfall events, and numerous melt

and refreeze events that affect the SWE.

The Arctic study site is located at 30 masl in the Trail Valley Creek research observatory (TVC) (Fig. 1) (68.4° N,

133.3° W), 50 km north of Inuvik, Northwest Territories. The TVC site is characterized by continuous permafrost with a

shallow active layer. It is dominated by Arctic tundra vegetation, with the ground cover consisting of a highly porous organic

layer and a large water storage capacity (Quinton and Marsh, 1999; Wrona, 2016). Patches of tall shrubs (birch, alder, and

willow) and black spruce trees are scattered across the tundra. Snow cover forms in October and persists until May, with few

or no melt periods over the winter. This snow cover is shallow in the wind-blown upland areas and deep snow drifts form on

lee hillslopes, along stream channels and lake edges, and in tall shrub patches (Marsh and Pomeroy, 1996).

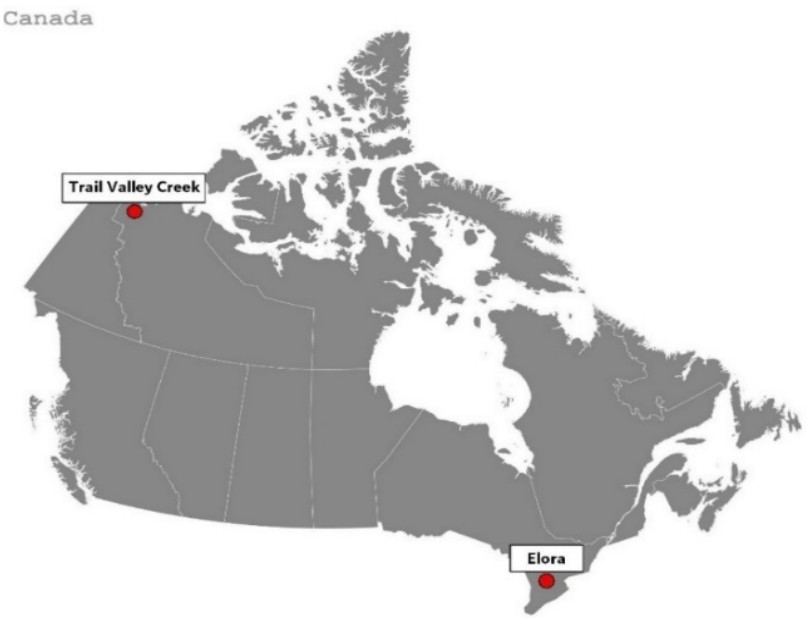

**Figure 1. Locations of the southern Canada (Elora) and western Canadian Arctic (Trail Valley Creek) sites used in this study. The**
**Elora site is located on an agricultural field and has a shallow, temperate snow cover. While the Trail Valley site is typical of the**
**tundra-taiga ecotone with snow that is highly variable in depth, density and SWE.**

**3.2 CRNS Installations**

A single CRNS was placed in the centre of the Elora field (Fig. 2). Installation of the CRNS occurred on February

11, 2017 for the 2016/2017 winter season, and on December 5, 2017 for the 2017/2018 winter season. The CRNS experienced

a power issue and did not record data from January 13 to 23, 2018.



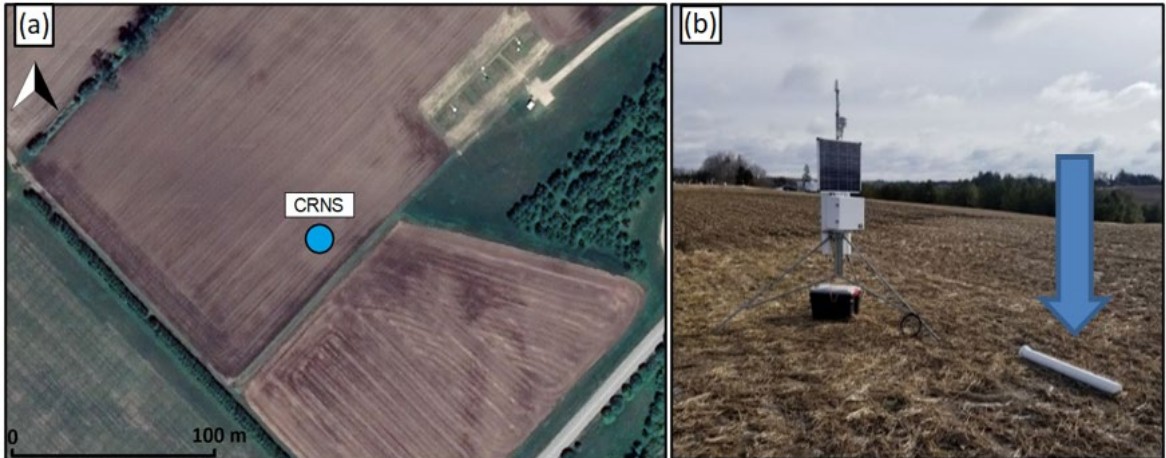

**Figure 2. (a) Location of CRNS at the Elora site during the 2016/17 and 2017/18 seasons (© Google Earth Pro 2020). (b) The location of the sensor tube is indicated by the blue arrow.**


Five CRNSs were installed at TVC on August 5, 2016 along a 50 m transect that traversed from tundra to alder shrubs (up to 2.5 m in height) and back to tundra. The CRNSs were installed concurrently, approximately eight meters apart (Fig. 3) and were connected to a single data logger. This shrub patch accumulates a deep snowdrift each winter that is representative of snow accumulation typical to shrub patches found in the tundra-taiga transition zone. Each CRNS was installed on the

ground surface prior to the accumulation of snow. The batteries for both the Elora and TVC systems were recharged by solar panels. However, at TVC, they provided limited power to the batteries during much of the winter. From the start of the TVC snow season in October, until March 4, 2017 and May 3, 2018, a low power sampling mode was used, with four, one-hour recordings obtained per day. After these dates, sufficient sun allowed the solar panels to recharge the batteries, and the CRNS system measurement frequency adjusted to 24, one-hour recordings per day. During the winter period, we used a 12-timestep

running average to estimate SWE; resulting in a three-day averaged SWE which was used in our analysis. After March 4, 2017 and May 3, 2018, we used a 12-hour running average SWE. The TVC CRNS system experienced a power failure from November 10 to 27, 2017, and as a result, no data is available for this period.





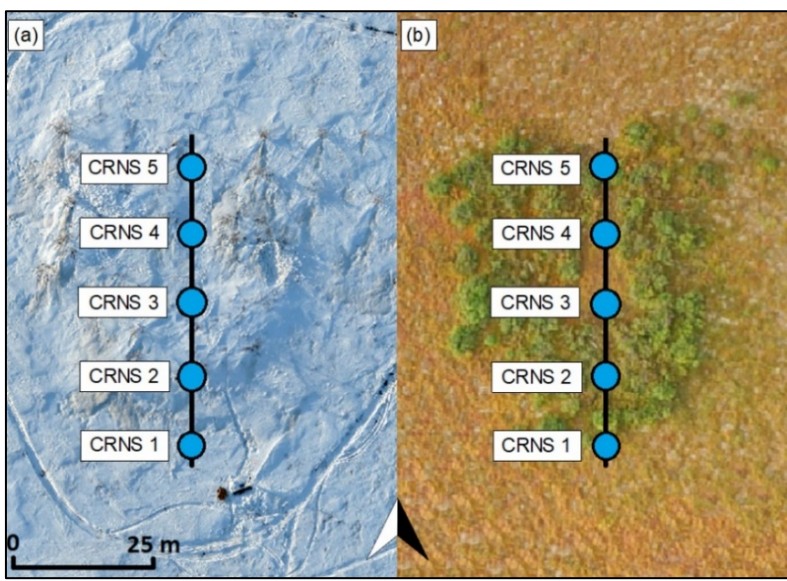

**Figure 3. CRNS transect at Trail Valley Creek during the 2016/17 and 2017/18 field seasons. (a) Site transect during winter sampling.**
**(b) Site during snow-free conditions, this image displays the tall alder shrub vegetation (green) and tundra vegetation (orange). Each sensor tube is the same size and style as shown in Fig. (2b).**

### 3.3 Snow Surveys

A total of five snow surveys were conducted at the Elora site during accumulation and melt conditions from February 11

to March 14, 2017, and 11 surveys from December 23, 2017 to February 20, 2018. Snow surveys used an ESC30 style snow-

corer from SnowHydro that features a cross sectional area of 30 cm$^2$ for measuring snow depth and density via snow cores.

The snow cores were transferred to a plastic bag and weighed on-site with an electronic scale (A&D HT-3000). The depth and

density of each snow sample was recorded and used to calculate the SWE. Snow surveys at this site consisted of three to four

snow core samples taken within a one meter proximity to the CRNS. Snow core results were averaged to represent a single

value for that date. Results from Turcan and Loijens (1975), Peterson and Brown (1975), Goodison et al. (1981), and Sturm et

al. (2010) state that the standard measurement error associated with using this type of snow-corer ranges from 1-10 %.

Snow surveys were performed at the TVC site during the 2016/2017 winter season from December 13, 2016 to June 6,

2017 and April 28, 2018 to June 7, 2018 during the 2017/2018 season. 17 surveys were conducted in 2016/17 and 28 in the

2017/18 winter season. The surveys included accumulation and snowmelt conditions and consisted of 10 measurements

(approximately equally spaced apart) along the 50 meter transect (Fig. 3). Again, a SnowHydro snow-corer was used. Using



the same approach as the Elora site, samples had their depth and weight recorded immediately after collection and were used to calculate SWE. Data from this site was used in two ways,

1) SWE calculated from the five CRNSs was averaged and this single value was used to represent the total snowdrift for that date, and

2) SWE calculated for each CRNS was compared with the snow survey measurement obtained nearest to the specific

CRNS of interest. This allowed the CRNSs to be compared to one-another within the snowdrift over the course of the snow covered season.

### 3.4 CRNS Parameters

The instrument manufacturer provided two sets of calibration parameters, used in Eq. (5), for the CRNS instrument. The $\Lambda_{max}$ value represents the rapid attenuation of neutrons, while the $\Lambda_{min}$ value represents a more gradual attenuation. $a_1$, $a_2$

and $a_3$ are fitting parameters determined by the manufacturer through calibration and field validation experiments (Howat et al., 2018; Gugerli et al., 2019). The standard terrestrial parameters (Table 1) were used for the Elora study site. However, the TVC snow cover is underlain by a high porosity soil matrix with an active layer thickness of 0.5 to 1.0 m (Wilcox et al., 2019), this active layer is typically saturated with liquid water prior to freeze up, and therefore has a high ice content during the winter season (Wrona, 2016). As a result, we applied the manufacturer suggested glacier parameters (Table 2) to this Arctic site.

Additionally, we increased the $a_1$ parameter in order to create a site-specific calibration which addressed the factor that the TVC subsurface was not pure water/ice, but had mineral and organic properties as well. Howat et al. (2018) and Gugerli et al. (2019) tested a similar CRNS model on the Greenland Ice sheet and the Glacier de la Plaine Morte in Switzerland, and were successful using the manufacturer-provided parameters. Although they tested a non-invasive CRNS model, findings from Schattan et al. (2017) and Schrön et al. (2017) suggest that adjusting the calibration parameters may lead to improved results,

and in discussion with the manufacturer, it was confirmed that adjusting the calibration parameters for this in-situ CRNS model may also lead to improved results.



**Table 1. Weighting function parameters for Eq. (5) were used for the Elora site. Values were obtained from the CRNS manufacturer and are representative of a terrestrial landscape.**

| Elora | |
|---|---|
| $\Lambda_{max}$ | 134.7 |
| $\Lambda_{min}$ | 20.0 |
| $a_1$ | 0.612 |
| $a_2$ | 0.073 |
| $a_3$ | 0.598 |


**Table 2. Weighting function parameters used for Eq. (5) for the Trail Valley Creek shrub site. Values were obtained from the CRNS manufacturer and are representative of a glacier landscape. $a_1$ parameter was adjusted from 0.313 to 0.355 to represent the high porosity, saturated soils of the study site.**

| Trail Valley Creek | |
|---|---|
| $\Lambda_{max}$ | 114.4 |
| $\Lambda_{min}$ | 14.1 |
| $a_1$ | 0.355 |
| $a_2$ | 0.083 |
| $a_3$ | 1.117 |

**4. Results and Discussion**

**4.1 Relationship between Neutron Counts and SWE**

Corrected, moderated neutron counts, N from Eq. (1) (simply referred to as counts, or neutron counts, for the remainder of the paper), were assessed in relation to SWE at both study sites as follows.

***Elora:*** The relationship between neutron counts and SWE at Elora was assessed in three ways. First, Eq. (4) was used to
estimate SWE from the neutron counts and compared to snow survey estimates of SWE. Using this approach, the $R^2$ was 0.79 when combining data from 2016 and 2017. Second, we carried out a bivariate analysis directly between neutron counts and SWE from the snow surveys using a linear regression (Fig. 4a). Using this approach provided a best fit linear regression that varied between each study year as follows:

$$\text{SWE}_{\text{Elora2016/17}} = -0.084(N) + 112.00 \tag{6}$$



and

$$SWE_{Elora2017/18} = -0.144(N) + 191.99 \qquad (7)$$


Where $N$ is the 12-h averaged counts (N) from Eq. (1) which has already been normalized by the snow-free near-surface

water content from Eq. (4). The RMSE of the CRNS-measured SWE was 1.8 mm in 2016/17 and 2.3 mm in 2017/18 and

Pearson correlation coefficients were -0.95 for 2016/17 and -0.97 for 2017/18. $R^2$ values were 0.92 and 0.94, respectively.

These values suggest a strong correlation to the linear regression equations and may indicate a high probability of predicting

future responses. However, the slope and y-intercept values for 2016 vs 2017 (Eq. (6) and (7)) are considerably different.

The discrepancy in the y-intercept is believed to be related to the CRNS being installed later in the winter season in 2016/17

(February 11, 2017) after the first accumulation of snow at the site. As a result, the sensors neutron count baseline does not

incorporate the near-surface water content prior to the initial winter freeze-up, and therefore, based on Eq. (4),

underrepresents the actual SWE at this site. The majority of this unaccounted water content is likely stored in the first few

centimeters of soil. In 2017/18 however, the CRNS was installed on December 4, 2017, before the first accumulation of

snow on the ground and the initial soil freeze-up. This means the CRNSs' baseline between the two seasons are likely to be

considerably different. To consider if this explanation is reasonable, we followed the approach of Sigouin and Si (2016),

where the authors estimated soil water storage in the top 10 cm of the soil profile and adjusted their SWE values accordingly.

To follow this approach, we used an estimated water capacity of the top soil layer to be 1.3 to 2 mm/cm (Blencowe, 1960;

Ball, 2001), and assumed a 50% soil moisture. This provided an estimated soil water storage of approximately five

millimeters. Adding this value to the non-zero SWE from snow surveys conducted in 2016/17 and conducting a second

regression to the 2016/17 data provided a best fit equation of:

$$SWE_{Elora2016/17adjusted} = -0.107(N) + 143.9 \qquad (8)$$


As shown in Fig. 4b, this adjusted equation provides a slope and y-intercept that is closer to that of the 2017/18 equation (Eq. (7)). The adjusted 2016/17 data yields an improved Pearson correlation coefficient of -0.98, an improved $R^2$ value of 0.96 and a slightly lower RMSE of 1.7 mm. This illustrates that it is extremely important to install CRNS prior to the start of the snow covered season.

A third approach to estimating SWE follows that of Kodama et al. (1979) to use an exponential relationship between SWE and neutron intensity. Prior to assessing an exponential curve, all zero values were changed to reflect non-zero values, as a result, these values were altered to 0.01 mm of SWE. An exponential regression was then performed and yielded an $R^2$ value of 0.55. Since this value is considerably lower than the two approaches noted above, it was not used for the remainder of this paper.




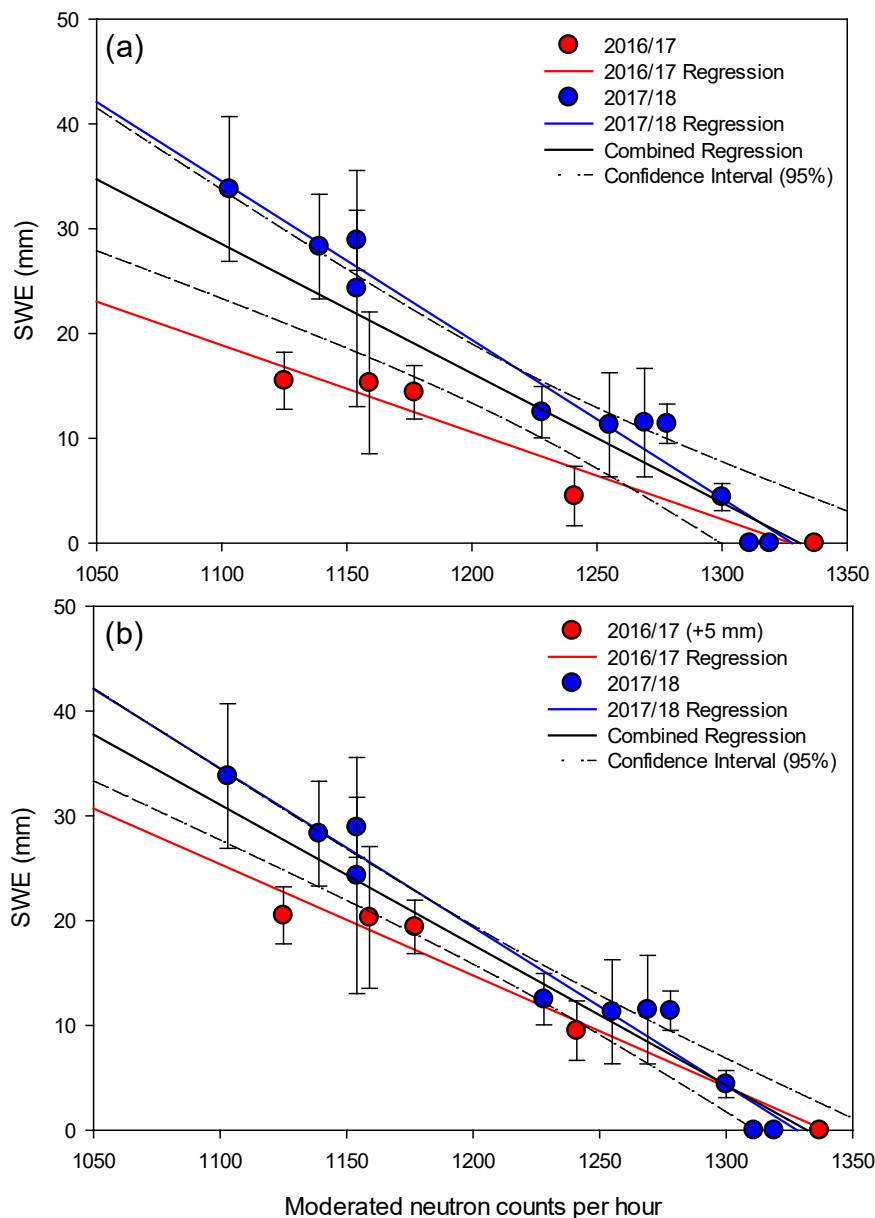

**Figure 4. Comparison between counts (N) and average snow survey SWE at the Elora site. (a) Shows both 2016/17 and 2017/18, with red and blue lines showing the regression Eq. (6) and (7). (b) Shows 2016/17 and 2017/18, but with the regression equation for 2016/17 SWE values adjusted to account for the antecedent water content in the top few centimeters of soil. Red line represents Eq. (7) and blue, Eq. (8). Zero SWE values represent snow-free conditions and the error bars represent the standard deviations.**



*__Trail Valley Creek__*: For TVC, we used the first two approaches outlined above to consider the relationship between counts and SWE. First, Eq. (4) was used to estimate the relationship between neutron counts and SWE from snow surveys. Using this

approach, the $R^2$ was 0.89 when combining data from 2016 and 2017. The second approach again directly compared neutron counts and SWE from the snow surveys using the same approach as above and shown on in Fig. 5. For 2016/17, the empirical equation was:

$$SWE_{TVC2016/17} = -0.679(N) + 515.1 \hspace{3cm} (9)$$


where $N$ is the 12-h averaged counts (N) (with the exception of snow survey measurements from December 13, 2016 and February 1, 2017 – which were three-day, 12-timestep averaged values) which has already been normalized by the snow-free near-surface water content factor from Eq. (4). The CRNS-measured SWE was similar to that of the snow surveys and had a RMSE of 39.6 mm. the Pearson correlation coefficient of Eq. (9) was determined to be -0.95, with an $R^2$ value of 0.91,

indicating that the majority of the variance was explained by the associated counts.

For the winter of 2017/18, the slope and intercept were similar to that of 2016/17, with the empirical equation as follows:

$$SWE_{TVC2017/18} = -0.715(N) + 500.9 \hspace{3cm} (10)$$


The Pearson correlation was determined to be -0.89 with an $R^2$ value of 0.79, and had an RMSE of 27.8 mm. When accounting for the error bars, all snow surveys conducted in the 2016/17 and 2017/18 winter season are within the 2016/17 and 2017/18 regressions. The similarity in the slope and y-intercept of Eq. (9) and (10) demonstrates the consistency of winter accumulation and melt at this site and suggests that a single regression equation could be used for future years. When combining the 2016/17

and 2017/18 regressions from TVC, the equation becomes:



$$SWE_{TVC-CombinedYears} = -0.654(N) + 492.0 \qquad (11)$$

Equation 11, the combined years regression, was determined to have a Pearson correlation of -0.95, an $R^2$ value of

0.87, and an RMSE of 34.2 mm. When considering Arctic, Antarctic, or other sites similar to TVC; where the annual SWE is

relatively consistent and does not experience mid-season melt and refreeze cycles, the annual regression formulas can be

combined to create a single regression equation, which can then be used to predict SWE for future years. Considering that the

in-situ CRNS requires virtually zero maintenance, the combined regression equation method can be a significant advantage

for SWE measurements at remote sites in comparison to other instrumentation.

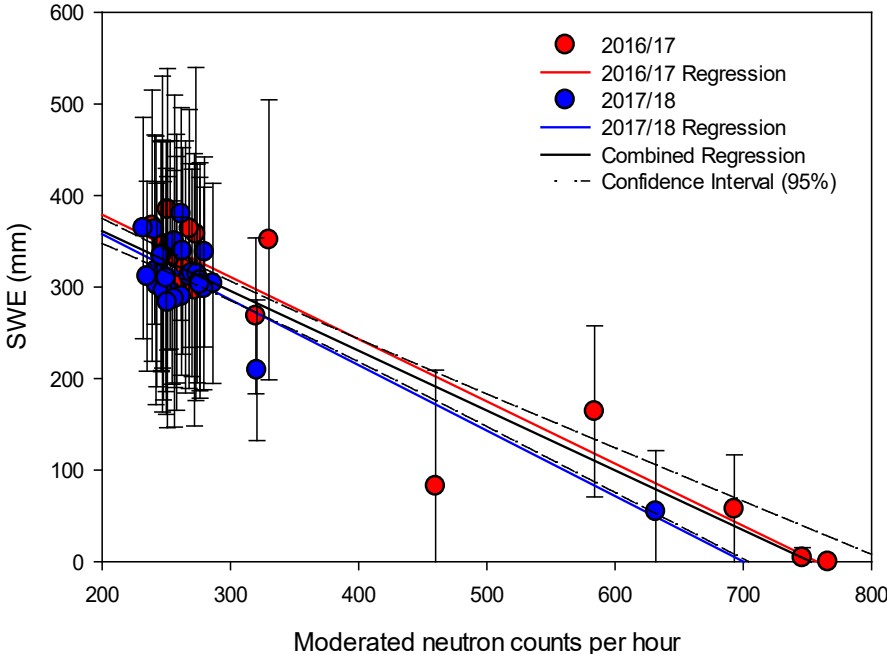


**Figure 5. Comparison between counts (N) and average snow survey SWE at the Trail Valley Creek snow drift site. Red and blue lines represent Eq. (9) and (10). Zero SWE values represent melt conditions (snow-free) and the error bars represent the standard deviations of each snow survey.**



## 335   4.2 Temporal Snow Cover Development and Melt

Using Eq. (4), or the relationships between neutron counts and SWE as described above (Eq. (7), (8), (9) and (10)), the CRNS instrument allowed for the continuous measurement of SWE over an entire winter accumulation and melt season at one point at the Elora site, and for multiple sites across the TVC snow drift. Figure 6 shows changes in SWE at the Elora site for both study years, and illustrates the potential for the CRNS approach to measure key aspects of the winter SWE, including:

maximum SWE, rapid changes in SWE due to both snowfall accumulation and snowmelt, and the timing of snowpack removal due to melt. For example, during the 2016/17 winter season the maximum SWE peaked briefly at 31 mm in mid-February and 42 mm in late January 2017/18 (Fig. 6), and then rapidly decreased over the next few days due to snowmelt. Measuring such rapid changes in SWE would be very challenging using manual snow survey measurements, and only a few other instruments, such as gamma snow sensors, allow this type of high-temporal resolution, point SWE observations. The CRNS also shows

that in 2016/17, the site became snow free numerous times over the winter (Fig. 6), and the snow cover was removed for the last time on March 14. In 2017/18 there was a continuous snow cover from December to late January, and the snow cover was then removed on February 20 and did not form again that winter. The small, short duration fluctuations in SWE in both years (Fig. 6) represent the periods of snowfall, snowmelt, sublimation, and wind erosion/transport. In addition, small fluctuations are likely also due to the inherent measurement error of the CRNS. This error has yet to be definitively quantified but is

assumed to average below 7% (Kodama et al., 1979; Howat et al., 2018; Gugerli et al., 2019). Figure 6, at some intervals, shows negative SWE during both winters, this implies that the CRNS is recording a higher number of counts ($N$) than was originally measured during its baseline ($N_0$). Meaning that the CRNS is sensing a lower amount of near-surface water content than was recorded at the start of the winter season. This is directly due to the CRNS fundamental measurement basis where any deviations from the baseline counting rate are inversely proportional to the amount of near-surface water content (Eq. (4)).

In these cases, the negative values imply that the snow has melted, infiltrated past the measurement scope of the CRNS, and therefore the immediate surrounding environment is drier than it was just before the onset of the winter season's first snowfall and initial soil freeze-up.


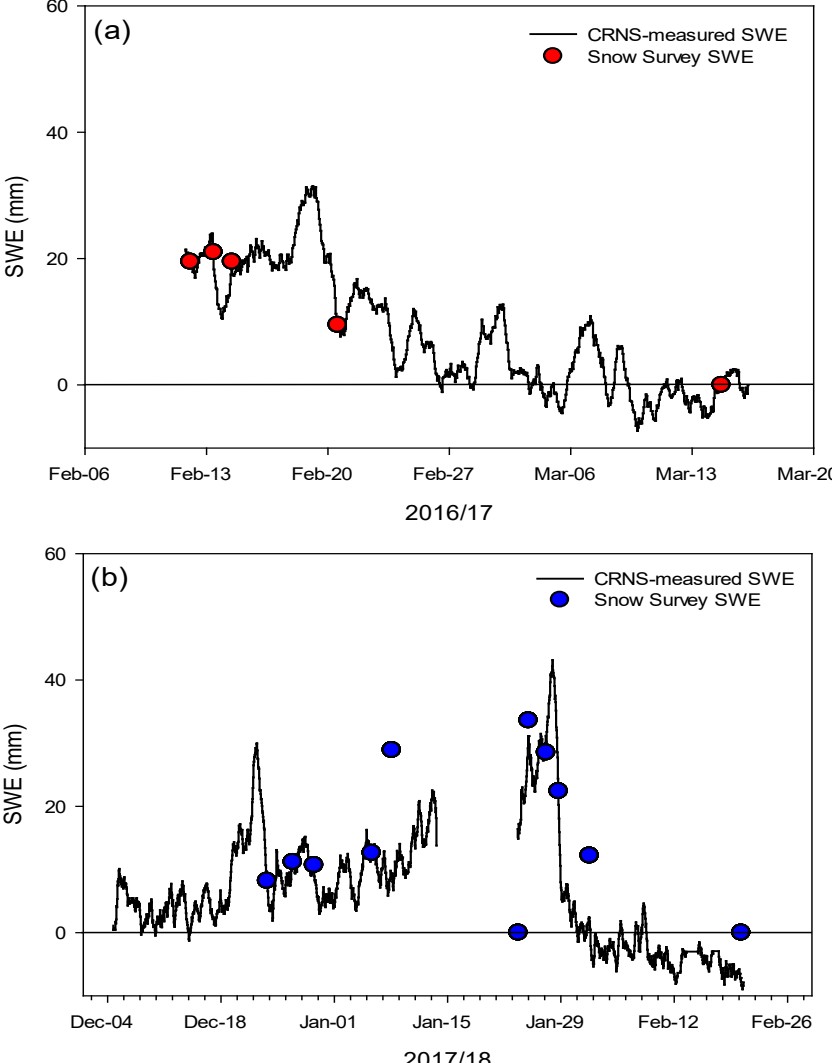

**Figure 6. Continuous measurement of SWE at the low-SWE, Elora site during the (a) 2016/17 and (b) 2017/18 winter season. When SWE values are negative, the CRNS is recording a lower near surface water content than its baseline. CRNS SWE values were calculated using Eq. (4).**

One example of the advantage of using a CRNS system is shown during 2017/18 (Fig. 6b) when there was a notable discrepancy on January 23, 2018 between the observed and estimated SWE. The CRNS estimated 16 mm of SWE, while the snow survey conducted on the same day resulted in a SWE of zero millimeters. This discrepancy occurred because a warm spell led to rapid snowmelt between January 21-22, immediately followed by a return to below freezing temperatures. This

resulted in the formation of a thick ice layer covering the site, which the snow survey was not able to measure. However, the CRNS was able to record the SWE of this ice layer.

Figure 7 shows a similar time series for SWE at the TVC site. In this example, the SWE from five CRNSs were estimated using Eq. (4) and compared to average snow survey data across the same transect. The initial snow-precipitation events of the 2016/17 season occurred in late November 2017, but a month earlier in 2017/18. During both years, SWE

continued to increase for the remainder of the winter. Unlike the Elora site, there were no midwinter melt events, but the small decreases in SWE are likely due to removal of snow from the transect by blowing snow erosion. The maximum average SWE across the transect in 2016/17 was 370 mm. Peak SWE occurred late in the season, one week prior to the onset of snowmelt. Small, high frequency SWE fluctuations during this period are primarily due to the change in the sampling rate of the CRNS. This change occurred when we switched the CRNS system from winter power conservation mode, where the sampling rate

was four, one-hour interval recordings per day, to the default sampling rate, which was 24, one-hour interval recordings per day. The maximum SWE at TVC in the 2017/18 season was 369 mm, and once again, occurred immediately prior to the initial onset of the spring snowmelt. Since the CRNS system measures total SWE (including liquid water within the snowpack), it does not identify when surface snowmelt begins, but instead detects when meltwater begins to leave the base of the snowpack and SWE begins to decline. This ability allows the direct measurement of snowmelt runoff from the snow cover and is an

extremely useful parameter for studying snowmelt runoff and for testing the performance of snow models used for modelling snowmelt runoff.

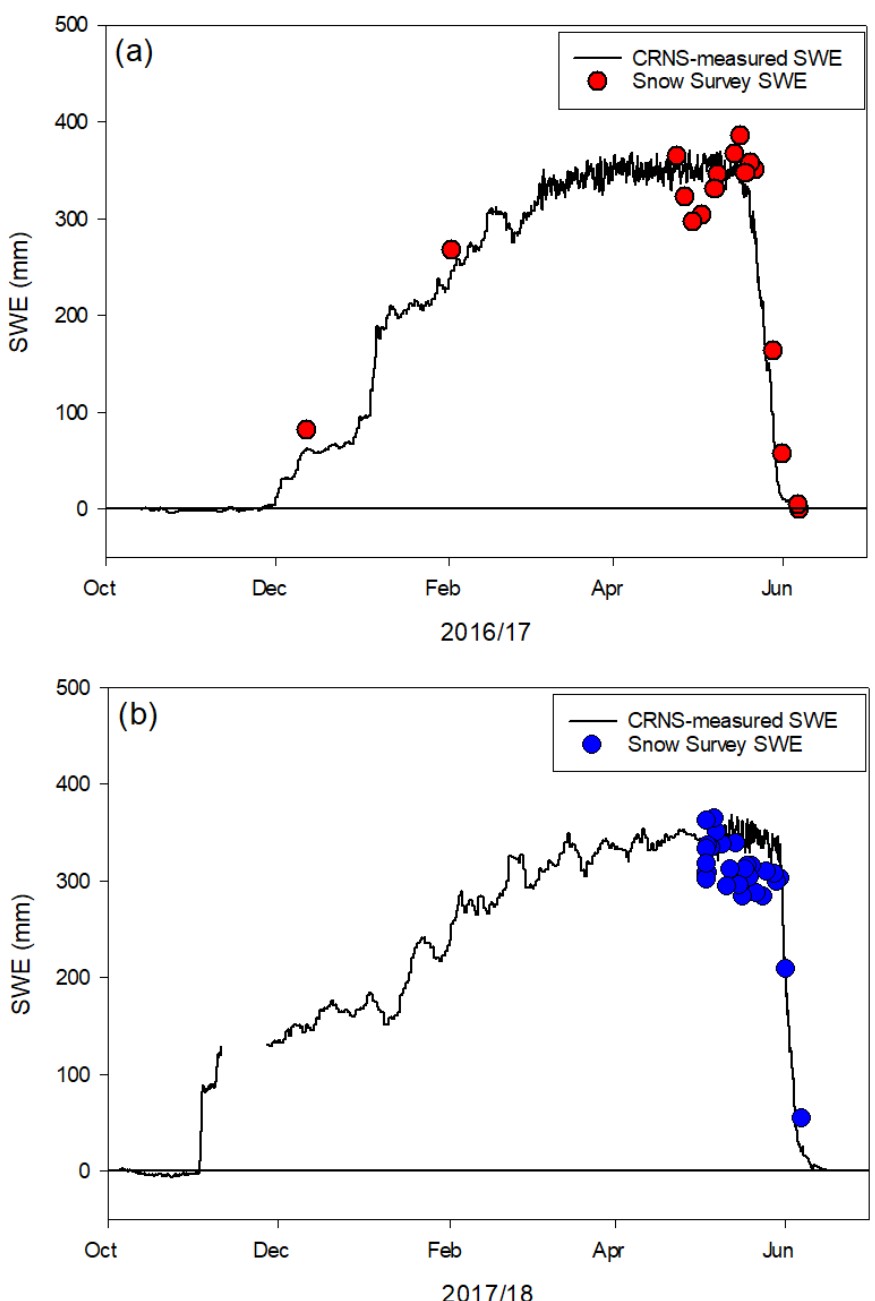

**Figure 7. Continuous measurement of a snowdrift at TVC in the (a) 2016/17 and (b) 2017/18 winter season. CRNS SWE values were calculated using Eq. (4).**






### 4.3 Snow Accumulation and Melt at locations across a Snow Drift

Figure 8 shows snow accumulation and melt at each of the five CRNSs across the TVC drift (Fig. 3). In the winter of 2016/17 (Fig. 8a), the snow drift began to form in the centre of the shrub patch in early December, while significant accumulation did not begin in the southern edge of the patch until a few weeks later. Over the rest of the winter, the snow

cover in the centre of the shrub patch (CRNS 3 and 4) continued to accumulate rapidly as blowing snow was deposited in the drift, and these sites ended up with the largest SWE at the end of winter. In this case, the center of the patch had 555 mm of SWE at the end of winter in 2016/17 and 645 mm at the end of winter in 2017/18. Other parts of the shrub patch also had similar maximum SWE values in comparison to one another from both years (Fig. 8a and 8b). As described earlier, the noisiness shown in Fig. 8 is due to a change in frequency of the sampling rate.

Spring snowmelt begins in mid-May at TVC, however, the early season melt is retained within the snowpack as liquid water is refrozen into ice (Pomeroy and Gray, 1995; Marsh and Pomeroy, 1996; Wrona, 2016). Further into the season, after sufficient melt, water is available to infiltrate the soil or runoff laterally (Quinton et al., 2010), exiting the measurement footprint of the in-situ CRNS and leading to the rapid decrease in SWE (Fig. 8a). Loss of SWE begins first at CRNS 1 (May 16), at the edge of the shrub patch where the snow is shallower. As melt progressed, snow mass is removed from each location

in the following order: CRNS 5 on May 17, CRNS 2 and 3 on May 20 and lastly, CRNS 4 on May 23. By June 7, all five CRNSs indicated that the snow overlying them has melted. In both seasons, CRNS 5 accumulated a higher SWE than CRNS 2 (Fig 8a and 8b) but began to melt days earlier. At the same time, as CRNS 5 is melting, CRNS 2, 4, and to a lesser extent, CRNS 3, experienced a slight increase in SWE, likely attributed to a lateral redistribution of snowmelt from the margin of the shrub site (CRNS 5) to the interior of the patch (CRNS 2, 3, 4). Using these changes in SWE from the CRNSs provides

continuous detailed snow accumulation and melt across a snowdrift for a complete winter season. This unique data set is particularly useful for large snowdrifts with considerable significance to their environmental landscapes and those that are vital for water resource management.



**Figure 8. Change in SWE at TVC from each individual CRNS throughout the (a) 2016/17, and (b) 2017/18 winter season. The locations of CRNS 1 to 5 are shown in Fig. 3. CRNS 1 and 5 are located at the edges of the shrub patch, and CRNS 2 to 4 are located in the centre of the patch. The sudden noisiness starting on March 4, 2017 in the 2016/17 season and May 3, 2018 in the 2017/18 season is due to the change in frequency of the sampling rate. This change in frequency occurred when we switched the CRNS system from winter power conservation mode, which had four, one-hour interval recordings per day, to standard power mode which had 24, one-hour interval recordings per day.**





## 4. Conclusion

In-situ CRNSs were tested at a temperate low-SWE agricultural field in Elora, Ontario and high-SWE Arctic tundra site in Trail Valley Creek, Northwest Territories. A strong negative correlation was found between the counts and the manual SWE measurements obtained from snow surveying at both sites (Pearson correlation values between -0.89 and -0.98). The
relationship implies that when SWE increases, the moderated neutron counts decrease. A unique advantage of CRNS systems is that ice layers and wet-snow from mid-winter melt events do not impact the sensor measurement accuracy and the CRNS measures all components of the snowpack SWE, including dry snow, ice layers, and wet snow. Using a CRNS for monitoring SWE provides a unique ability to continuously measure SWE and these systems can be installed in remote locations and in areas where performing regularly scheduled manual measurements are costly and logistically impractical. As such, the CRNS
system replaces the need of manually conducting snow surveys and requires virtually zero operational maintenance.

Since it was found that soil water in the top soil profile directly surrounding the CRNS affected the neutron intensity, future research involving a CRNS should examine at what soil depth the CRNS is impacted by soil water content, or alternatively, should be installed so that meltwater infiltration is shallow enough that water does not infiltrate past the base of the sensor. Additionally, we noted that the terrestrial set of parameters appeared to record low-SWE environments
exceptionally well, however, the glacier set of calibration parameters appeared to have some flexibility. This seems to indicate that site specific calibration may not apply only to the conventional parameters, such as the snow-free moderated neutron count ($N_0$), but also for the CRNS calibration function itself. SWE data from the CRNS could be used for validating surface mass balance models, verifying remote sensing approaches, and to better understand the effects of a changing climate on snowfall, mid-winter thaws, blowing snow, expanding shrubs capturing blowing snow, spatial variability in snow depth and snow water
equivalent (SWE), and the rate of spring melt – all of which are poorly known.

**Data Availability**

*Doi URL: Incoming – Placeholder.*

**Author Contributions**

This paper is the result of cooperation and collaboration with the listed co-authors. Dr. Philip Marsh is the principal co-author.

Dr. Marsh assisted with the design of the sampling plan, design of the manuscript, and provided edits and expertise throughout

the research process. Branden Walker reviewed the manuscript and provided support related to logistics and fieldwork. Darin

Desilets assisted with troubleshooting the CRNS instruments and formulated the instrument weighting function parameters.

**Competing Interests**

Dr. Philip Marsh is an editor for the peer-reviewed journal The Cryosphere.

**Disclaimer**

Any and all references made within this works to specific commercial products, services, manufacturers, trademarks, or

otherwise, does not imply its recommendation or endorsement by the authors.


**Acknowledgments**

Funding and logistical support for this study was provided by the Canada Research Chairs Program, Wilfrid Laurier University,

Natural Science and Engineering Research Council, Polar Knowledge Canada, Arctic Net, and the Polar Continental Shelf

Program. The authors sincerely thank Matthew Tsui, Dr. Barun Majumder, Brampton Dakin, Philip Mann and Dr. Aaron Berg.




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
