# Peer review of "Snow Water Equivalent Measurement in the Arctic based on Cosmicray Neutron Attenuation"

_The Cryosphere, 2021_

## Author Response (AR1)

**List of Changes:**

- Updated terminology of CRNS system to grounded in-situ.
- Added additional references/citations throughout the manuscript.
- Switched the positioning of section 3.3. and section 3.4. Also adjusted the positioning of text in the manuscript to where they are more appropriate (e.g. factory-fitting calibration to the end of Section 2.2.)
- Clarified the benefits and capabilities of the linear regression approach.
- Expanded on Section 4.3.
- Provided clarification why non-representative locations are often used for Arctic research bases.
- Updated the title of the manuscript.
- Clarified specialist terminology throughout the manuscript.
- Clarified that the snow surveys consisted of a snow-core campaign.
- Updated the terminology of the weighting function parameters to factory-fitting parameters.
- Clarified that a systematic approach was used to adjust the factory-fitting parameters and that data quality appeared to only increase when adjusting the a1 parameter.
- Clarified our assumption and how we obtained a 5 mm antecedent soil moisture value.
- Clarified that a late season installation of the CRNS system is still viable so long as a correction of soil moisture is applied – included new citation as support.
- Added a comparison from snow-core SWE to CRNS-estimated SWE at Elora at the start of Section 4.1.
- Created statistical summary table in section 4.1.
- Adjusted RMSE to be related to the maximum SWE value.
- Removed TVC linear regression from Section 4.1.
- Ensured figure error bars do not go past zero at any point.
- Clarified that the SWE signal in Figure 7 is averaged across the 5 CRNS in the transect – this is clarified in the Figure 7 caption, in text in Section 4.2. and for point "1)" in Section 3.4.
- Provided additional clarification for section 4.3. Adjusted the text wording to clarify that the interpretations are our own, and are based on established principles (snowmelt, infiltration, spring melt).
- Included 2017/18 temperature data in Appendix B to support that TVC experienced temperatures above and below freezing during the spring melt which likely lead to melt and refreeze events within the snowpack.
- Clarified the Peak SWE dates for TVC in Section 4.3.
- Shortened/updated figure/table captions.
- Updated conclusion section.

**Anonymous Reviewer 1 Responses:**

**Specific Comments**

L 19, L 21, L 68: I would recommend to write "buried CRNS" instead of "in-situ" as both systems are in-situ, depending on definition.

**Author reply:** The system terminology has been clarified and is now referred to as "grounded in-situ".

L 21 and throughout the manuscript: CRNS is often associated with the non-invasive application. I would thus use a different acronym like bCRNS for "buried CRNS" or similar.

**Author reply:** The system terminology has been clarified and is now referred to as "grounded in-situ".

L 51: add a reference for gamma attenuation as Koch et al is only about GNSS not gamma ray.

**Author reply:** A reference for gamma rays has been added.

(Kirkham, D., Koch, I., Saloranta, T., Litt, M., Stigter, E., Møen, K., Thapa, A., Kjetil, M., and Immerzeel, W.: Near Real-Time Measurement of Snow Water Equivalent in the Nepal Himalayas, Front. Earth Sci. 7: 177. Doi: 10.3389/feart.2019.00177.

L 65: Name the main technique, LiDAR (airborne and terrestrial) and its main obstacle: it is a campaign based measurement.

**Author reply:** The main technique and its primary obstacle have been named.

L 75ff: There is at least one more manufacturer of commercial CRNS systems (Geonor). But maybe it would be better to focus on the scientific usage of the measurement principle: Japan (Kodama), France (EDF), glaciers (Howat and Guguerli). Also this type of instruments has been used in the USSR in the 1980ies. I miss one recent application for shallow snow packs in the UK (https://onlinelibrary.wiley.com/doi/full/10.1002/hyp.14048) where they also found that adaptation of parameters is necessary.

**Author reply:** We have decided to focus on the scientific usage and have maintained the originally listed manufacturers (EDF, Hydroinnova) and applications (Kodama, EDF, Howat and Guguerli). The reference Wallbank et al., 2021 is included and we've noted that although they used a different type of CRNS system, they also found it necessary to adjust the parameters.

L 86: As CRS-1000 and SF are commercial names I would rather refer to the measurement principle (above-snow, buried).

**Author reply:** All references to the CRS-1000 are removed. There are a few references of the "SnowFox" included at the end of Section 1 as this is a modern CRNS with limited testing for SWE applications. Because of this, we believe there is value in identifying it. This is consistent with other publications from the Cryosphere which also utilized this model.

Howat et al. (2018); https://tc.copernicus.org/articles/12/2099/2018/ and Gugerli et al. (2019); https://tc.copernicus.org/articles/13/3413/2019/.

From Section 2 onwards, the system is strictly referred to as a grounded in-situ CRNS.

Chapter 3.1 and 3.4: I would recommend merging these two into one chapter, or alternatively switching 3.3 and 3.4 to not confuse the reader.

**Author reply:** Section 3.3. and section 3.4. are repositioned/switched.

Chapter 4.1: What can we learn from these regressions? It would be necessary to discuss the added value and the implications of this analysis.

**Author reply:** Text has been added to clarify the benefit of the linear regressions. In summary, the linear regressions are well transferable in time at sites where soil moisture conditions are known and/or rather consistent. The linear regression equations can also be used to estimate the soil water storage. Using a linear regression equation is significantly more time efficient than fitting the full N0-calibration function.

Chapter 4.3: This is a very interesting part of the paper and it should be more pronounced, especially as compared to the linear regression analysis, as it comprises the main novelty of the paper.

**Author reply:** Clarification has been added to this section.

L 431: Please write more clearly where the influence of the top soil profile was found, and also add it more clearly in the discussion section.

**Author reply:** We've clarified our assumption. The estimated water capacity is in the top 10 cm of the soil layer and up to 2 mm/cm (Blencowe, 1960; Ball, 2001). We assumed a 50% soil moisture on the top 10 cm of soil and this provided an estimated soil water storage of up to 5 mm. A similar approach was used by Sigouin and Si (2016).

**Anonymous Reviewer 2 Responses:**

**Detailed comments/questions:**

**Title:**

I see two issues in the proposed title: 1) "Cosmic-ray" is a physical process on which the measurement method is based, not a method by itself, and 2) the proper physical variable measured here is the SWE, which deserves to be mentioned in the title. An alternative proposition for the title could be: "Snow Water Equivalent measurement in Arctic based on Cosmic-ray neutron attenuation".

**Author reply:** The title has been updated to "Snow Water Equivalent measurement in the Arctic based on Cosmic-ray Neutron Attenuation"

**Abstract:**

Lines 23-25 the correlation (both Pearson and $R^2$) and RMSE values provided here are not very informative in an abstract, especially considering the limits of the use of linear regression in this context (see comments below). A more qualitative indication on the accuracy/reliability of such sensors in the Artic context is expected.

**Author reply:** Abstract has been updated.

**Text:**

Line 35: "often less than 300 mm": is it snow height or SWE?

**Author reply:** This is clarified to note "total SWE is often less than 300 mm".

Line 45: "often located at non-representative locations": what does it mean? Why such locations have been chosen then?

**Author reply:** We've clarified what this means and why non-representative locations are commonly chosen.

Lines 51-52: "using gamma attenuation […] but again are limited to point measurement": as mentioned line 65, airborne gamma methods can provide snow mapping, thus are not limited to point applications.

**Author reply:** Clarification is provided to note that gamma attenuation methods are limited to point or campaign-based measurements and have limited applicability in the Arctic due to the high cost associated with campaign-based measurements.

Line 67: "Cosmic ray attenuation methods have not been extensively tested": this is not true. Several wide-scale fleets of CRNS are currently in operation, for industrial use, e.g. in France (about 40 sites, the oldest since about 20 years now, see Paquet, E., & Laval, M. T., 2006) or Spain (Cobos et al., 2010). The French example is anyway evoked later in the manuscript… Bogena et al. (2021) indicate that

"worldwide, about 200 stationary Cosmic Ray Neutron Probes have been installed since the introduction of the method".

**Author reply:** Clarification is provided throughout the text that grounded in-situ CRNS, which are not buried, and for use in the Arctic, have only had limited testing.

Line 70: "neutrons in the fast to epithermal range": please define this terms for non-specialists.

**Author reply:** We have defined these terms by providing their energy range.

Line 134: "neutron count during the time of interest": please define this time of interest.

**Author reply:** The time of interest is clarified as the snow covered season.

Line 154: please provide the reference in which this attenuation coefficient has been introduced, and the physical/statistical processes to which the parameters are linked.

**Author reply:** The attenuation coefficients were provided by the manufacturer and determined through field validation and calibration studies. This is noted at the end of section 2.2.

Line 213: "the surveys included accumulation and snowmelt conditions": was it more than a snow-core campaign? If not, the term "snow-core measurement" is more appropriated.

**Author reply:** We removed "..included accumulation and snowmelt conditions" and updated the text to note that the "snow surveys consisted of a snow-core campaign".

Lines 223-226: this part deserves to be moved at the end of the Part 2, just after Equation 5, as it provides details about the calibration parameters used in this equation.

**Author reply:** This part has been moved to the end of Section 2, just after Eq. 5.

Line 230: "Additionally, we increased the a1 parameter in order to create a site-specific calibration":  why this parameter? Given the equation 5, it adds/removes counts from the corrected N value, accounting for attenuation not related to SWE.

**Author reply:** Our approach has been clarified – we used a systematic approach on each of the parameters and observed a significant increase in data quality, in comparison to field measurements, when adjusting only the a1 parameter. We have suggested for future research to investigate the impact of each parameter and to explore the identification of a standard set of factory-fitting parameters for a typical Arctic landscape.

Tables 1&2: the captions refer to "weighting function parameters" of the Equation 5, although this equation describes an "attenuation coefficient" (line 152).

**Author reply:** The parameter terminology has been adjusted to "factory-fitting parameters".

Lines 251-252: "a bivariate analysis [...] using a linear regression": I think this is the main issue of this study. Why using a linear model considering the highly non-linear link between N and SWE, given Equations 4 and 5? Furthermore, a detailed formulation of the attenuation coefficient has been introduced just before, allowing a local calibration of its parameters. Why not using it? If it is useless or difficult here, thus leading to a simpler model, this deserves to be detailed and justified.

**Author reply:** Text has been added to clarify the benefit of the linear regressions. In summary, the linear regressions are well transferable in time at sites where soil moisture conditions are known and/or rather consistent. The linear regression equations can also be used to estimate the soil water storage. Additionally, using a linear regression equation is significantly more time efficient than fitting the full N0-calibration function.

The non-linear approach was solely utilized in sections 4.2 and 4.3. No changes to the manuscript in this regard.

Lines 255-284: this part is not easy to read, the numerous values and statistical scores provided deserve to be gathered in a table. The correlation coefficients R, both Pearson and $R^2$, somehow rely on the hypothesis of a linear relation between the two variables, which is problematic here, as written above. The RMSE should be related to the maximum SWE value to be informative. As written above, one expects a comparison with the no linear model SWE=f(N) based on equations 4 and 5.

**Author reply:** A statistical summary table is included (Table 3). A brief comparison between the non-linear model and snow surveys has been included at the start of Section 4.1. The correlation coefficients are relevant as they indicate a strong negative relationship between the moderated neutron counts and the manual SWE measurements and suggest that the majority of the variance is accounted for. For this reason, we kept the R and $R^2$ values in text.

Lines 274-277: Once again, I don't understand why such an "expert" correction (which is not of second order here, 5 mm have been added to SWE of about 15-35 mm) has to be introduced before inferring a linear model, whereas such an effect could be compensated thanks to parameter a1 in Equation 5.

**Author reply:** We've clarified our assumption. The estimated water capacity is in the top 10 cm of the soil layer and up to 2 mm/cm (Blencowe, 1960; Ball, 2001). We assumed a 50% soil moisture on the top 10 cm of soil and this provided an estimated soil water storage of up to 5 mm. Adjusting the a1 parameter would impact all values while incorporating 5 mm of the estimated soil water storage only impacts non-zero SWE values. A similar approach was used by Sigouin and Si (2016).

Lines 283-284: "it is extremely important to install CNRS prior to the start of the snow covered season": considering this, I am not sure that the Feb-Mar 2017 data at Elora are usable in this study.

**Author reply:** The late season installation allowed us to reasonably estimate the antecedent soil water capacity by comparing the regression trendlines from year-to-year. This approach has not been documented in literature and may serve researchers and operators by essentially validating a late-season setup – so long as it is adjusted using a full-season dataset from the same site. This is clarified in Section 4.1.

Lines 285-289: I wonder why the Kodama approach (somehow the "father" of the cosmic-ray based SWE measurement) is not appropriated here, but neither the model used nor the comparison to snow core SWE are provided to illustrate this issue.

Author reply: A comparison to snow-core SWE for Elora has been included at the start of Section 4.1.

Lines 298-329: Same remarks as for Elora results. Furthermore, the data shown in Figure 5 look like a typical pitfall for linear regression, with most of the data grouped between N values of 200-300, and few others above 600. At least a Theil-Sen regression could have been used for a more robust estimate. It is not clear whether the confidence interval on the SWE values of the Figure 5 is the one of the snow core measurement. In that case, for low SWE values, the lower part of this confidence interval goes to negative SWE, with is not realistic for a snow core measurement (at worst, no snow is cored).

Author reply: The TVC linear regression was removed in Section 4.1. After discussion, we decided to elaborate on the linear regression approach rather than incorporating a new regression, such as a Thiel-Sen. The error bars have been removed where applicable so they do not go past zero SWE.

Line 334: "Using Equation 4, or the relationships between neutrons counts and SWE": according to the caption of the Figures 6 and 7, the continuous SWE signal is computed thanks to the non-linear relation given in Equation 4. Then what is then the point of the linear models presented before?

Author reply: The advantages of the linear regression have been clarified. We've also noted the manufacturer statement that the linear approximation may be effectively utilized for up to 15 cm of SWE, as the initial part of the $N_0$-calibration function is fairly linear and only past 15 cm does the non-linearity of the function become more prominent.

Line 348: "snowfall, snowmelt, sublimation and wind erosion/transport": at this point, these are only guesses of the involved processes. The CRNS measurements should be completed at least by wind and temperature measurement to confirm this, providing proxies.

Author reply: We do not have the applicable data. I've noted that the small, short duration fluctuations in SWE in both years "likely" represent the periods of snowfall, snowmelt, sublimation, and wind erosion/transport.

Line 372: Does this mean that the SWE signal presented in the Figure 7 is averaged across the 5 CRNS in the transect?

Author reply: Caption in Figure 7 was updated to clarify that the CRNS-measured SWE was averaged from the 5 CRNS. Additional clarification included in Section 4.2. Point "1)" in Section 3.4 was also updated to clarify that SWE calculated from the five CRNS instruments was averaged and this single averaged value was used to represent the total snowdrift for that date.

Line 377: "Peak SWE occurred [...] one week prior to the onset of the snowmelt": Considering the "noise" of the SWE signal, the maximum snowpack seems to be reached almost one month before. Same remarks applies to line 381.

Author reply: The peak SWE dates have been clarified and included into the text in Section 4.2.

Line 392-409. I felt uncomfortable to see some direct interpretations of the presented data mixed with some other considerations drawn out form literature and not directly deducible from the observations, like in lines 400 to 404. This is somehow an over-interpretation of the presented results. The spatial pattern of the snowdrift could be better illustrated by plotting the ratio of local SWE v/s the averaged value, which could show that, at least for the 2016/2017 season, the spatial pattern of the snowdrift is rather stable throughout the season.

**Author reply:** Additional clarification is provided. The positioning of references has been adjusted. We do not agree that the spatial pattern is better illustrated by "plotting the ratio of local SWE v/s the averaged value".

Line 410-412: "This unique dataset […] water resource management": this conclusion is somehow quite emphatic considering the results presented, interesting but limited in time and space..

**Author reply:** Although we maintain our original position, we have expanded the text.

Figure 8: The caption is too long and too detailed, some of the details are (or could be) given in the text.

**Author reply:** The caption has been shortened; details that are provided in the text were removed from the caption.

Line 423: "A strong negative correlation was found between then counts and the manual SWE measurements": given the principle of measure of the CRNS, the contrary would have been a great surprise. I am not sure it is significant conclusion to be put here.

**Author reply:** We maintain this statement as if a weak negative correlation was found it may indicate that our snow survey values were erroneous or had some sort of discrepancy – in this works, this was not the case and so we maintain to include this statement.

Line 432: "the terrestrial set of parameters […] however, the glaciar sets of parameters": please refer to the tables 1 & 2. Once again, the conclusions here, like the chapter before, deal with the "non-linear" formulation whereas the "linear models" have been extensively documented in the paragraph 4.1, but finally poorly used in the study. A detailed and illustrated scoring of the "non–linear" formulation deserves to be presented instead.

**Author reply:** The conclusion summarizes the primary findings from all sections.

Line 438-440: Same remark as for lines 410-412.

**Author reply:** The conclusion summarizes the primary findings from all sections.

**Community reviewer responses:**

But, it is a rather technical article on the evaluation of a CRNS sensor, in that sense it is generally not the type of article from The Crysopshere?

**Author reply:** No changes to the manuscript were applicable. Note the similar papers with a technical focus published on TC. Howat et al (2018) https://tc.copernicus.org/articles/12/2099/2018/ and Sigouin and Si (2016) https://tc.copernicus.org/articles/10/1181/2016/tc-10-1181-2016.html

Unfortunately, one regrets the comparison of the SnowFox with the CRS-1000 which was also installed at TVC, I think?

**Author reply:** This works solely focuses on the SnowFox CRNS model due to its limited testing in an Arctic landscape.

The weak point of this sensor raised about the need for soil water calibration is clearly highlighted. The authors could refer to the sensor comparison paper:

Royer A., A. Roy, S. Jutras and A. Langlois (2021). Review article: Performance assessment of electromagnetic wave-based field sensors for SWE monitoring. The Cryosphere Discuss. [preprint], https://doi.org/10.5194/tc-2021-163, in review, 2021.

**Author reply:** This source consisted of relevant information and was included as a citation throughout the revised manuscript.

The continuous SWE measurement from 5 sensors along a transect (even a short one: 5 m) is new and interesting. The comparisons of the 5 sensors in Fig. 8 clearly show the spatial variability along the transect. This also shows the difficulty of linking in-situ measurements to the CRNS measurements.

**Author reply:** No applicable changes to the manuscript.

Have you tested the intercalibration between the 5 sensors?

**Author reply:** No applicable changes to the manuscript.

We would have liked to see a figure showing these spatial variations for some dates, in relation to the height of the shrubs for example. Is the sensor on the edge really representative of the tundra (it seems close to the shrubs, as seen in Fig. 3)? An analysis of the evolution of snow heights and densities along the transect would have been very interesting.

**Author reply:** Clarification is provided – the transect ranged from a tundra-shrub interface to alder shrubs and back to a tundra-shrub interface.

**Specific comments (L = line in the pdf version on line)**

L49-50 The statement " Measurements of snow depth are typically not representative of the surrounding natural terrain as they are limited to point observations using ruler measurements or acoustic distance systems " is also valid for SWE.

**Author reply:** This is among the reasons we focused the study on an Arctic snowdrift. Although deep drifts are small in area, they often contain a large portion of the total landscape SWE (Gray et al., 1974; Marsh and Woo, 1981; Gray et al. 1989; Marsh and Pomeroy, 1996; Sturm et al., 2001). No applicable changes to the manuscript.

L87 "the SF measuring point SWE along a transect."  specify: with several instruments. As written it is not clear.

**Author reply:** This has been clarified: "…the SF is capable of measuring SWE across deep snow drifts by employing multiple instruments in a transect."

L 175 The map in Fig. 1 does not seem to me to be very useful or necessary?

**Author reply:** There is value in immediately identifying the general location of the project sites – particularly for readers that are not knowledgeable on Canadian geography. No applicable changes to the manuscript.

L210 "standard measurement error » for snow core: see also the discussion in Royer et al. 2021 TCD.

**Author reply:** This has been included as a supporting citation in Section 3.4.

L245 Result section: Accuracy statistics should also be given relative to the SWE average, which is very low at the Ontario site (between 0 and 40 mm), and quite low at the TVC site (0-400 mm).

**Author reply:** This has been updated; however, accuracy statistics were given relative to the SWE maximum (as suggested by Anonymous Reviewer #2).

Unfortunately, the distribution of the measurement points at TVC results in a regression (SWE vs. Counts) being defined almost by two points: a "0" point and a mean  ~350 mm SWE point.

**Author reply:** The TVC regression has been removed from Section 4.1.

And why no scatterplot between SWE CRNS and SWE snow core?

**Author reply:** A scatterplot between CRNS SWE and Snow Core SWE for Elora at 2016/17 and 2017/18 has been included at the start of Section 4.1.

L421 Conclusion: Finally, what is the recommended sampling frequency?

**Author reply:** This is dependent on several factors. Some factors include power capabilities, application purpose and scope, prominence of the snow feature being monitored, length of observation/study, the weather conditions during the monitoring period, and the geographic location. The sampling frequency should be considered on a case-by-case basis. No applicable changes to the manuscript.

Would it not be advisable to shield the sensor to avoid the important problem of contamination of the SWE signal by soil moisture?

**Author reply:** More research needs to be done to extensively compare the results of a grounded-in-situ CRNS, which is buried, to those which are placed on the soil surface. This study focuses on the latter, and may be used in future comparison studies. Kodama et al. (1979) suggested either approach is feasible – we also believe both approaches have value. No applicable changes to the manuscript.

---

## Author Response (AR2)

**List of Changes:**

- Abstract (line 15): '..unattended measurements of snow water equivalent over..'
- Removed LIDAR, clarified that airborne radar methods require some knowledge of snow microstructure.
- Included (Derksen et al., 2017) citation in the References (line 516-518)
- Updated 'SnowHydro' to 'Snow-Hydro'
- Line 243 'Data…was used..' to 'Data…were used…'
- Lines 250-254 updated to note that the Elora dataset was tested according to Eq. (4) and additional linear regression, whereas the TVC dataset SWE was derived utilizing Eq. (4) only because the deeper snow there means the linearity between SWE and neutron counts no longer holds.
- Lines 264, 309, 311 changed 15cm to 150mm for consistency.
- Line 425 added '..but limited to SWE < 150mm' to the end of the sentence.